# Measuring what matters: Context-specific indicators for assessing immunisation performance in Pacific Island Countries and Areas

Cyra Patel[1]*, Ginny M. Sargent[1], Adeline Tinessia[2], Helen Mayfield[3], Dan Chateau[1], Akeem Ali[4], Ilisapeci Tuibeqa[5], Meru Sheel[1,2,6]*

**1** National Centre for Epidemiology and Population Health, Australian National University, Canberra, Australian Capital Territory, Australia, **2** Sydney School of Public Health, Faculty of Medicine and Health, The University of Sydney, Camperdown, New South Wales, Australia, **3** UQ Centre for Clinical Research, The University of Queensland, Brisbane, Queensland, Australia, **4** World Health Organization, Seoul, Republic of Korea, **5** Department of Paediatrics, Colonial War Memorial Hospital, Suva, Fiji, **6** Sydney Institute for Infectious Diseases, Faculty of Medicine and Health, The University of Sydney, Westmead, New South Wales, Australia

* Cyra.Patel@anu.edu.au (CP); Meru.Sheel@sydney.edu.au (MS)

## Abstract

Increasing countries' access to data can improve immunisation coverage through evidence-based decision-making. However, data collection and reporting is resource-intensive, so needs to be pragmatic, especially in low-and-middle-income countries. We aimed to identify which indicators are most important for measuring, and improving, national immunisation performance in Pacific Island Countries (PICs). We conducted an expert elicitation study, asking 13 experts involved in delivering immunisation programs, decision-makers, health information specialists, and global development partners across PICs to rate 41 indicators based on their knowledge of the feasibility and relevance of each indicator. We also asked experts their preferences for indicators to be retained or removed from a list of indicators for PICs. Experts participated in two rating rounds, with a discussion on the reasons for ratings before the second round. We calculated mean scores for feasibility and relevance, and ranked indicators based on experts' preferences and mean scores. We used framework analysis to identify reasons for selecting indicators. Experts agreed that certain indicators were essential to measure (e.g. data use in program planning and measles vaccination coverage), but preferences varied for most indicators. Preferences to include indicators in a set of indicators for PICs moderately correlated with scores for relevance (r = 0.68) and feasibility (r = 0.56). In discussions, experts highlighted usefulness for decision-making and ease of data collection, reporting and interpretation as the main reasons driving indicator selection. Country-specific factors such as health system factors, roles and influence of various immunisation actors, and macro-level factors (namely population size, distribution and mobility) affected relevance and feasibility, leading us to conclude that a single set of indicators for all PICs is inappropriate. Rather than having a strict set of indicators that all countries must

**Data Availability Statement:** Aggregated data from this study are reported in the results tables, with additional detailed findings reported in the supplementary materials. Interested readers may request that these same data compiled in a single Excel spreadsheet from the corresponding authors (Cyra.Patel@anu.edu.au or Meru.Sheel@sydney.edu.au). Individual-level data cannot be released as per the conditions of ethical approval and to which experts agreed when consenting to participate in this study. The conditions of this study were reviewed and approved by the Australian National University's Human Research Ethics Committee (Protocol 2022/368). Any requests for data is not already available in the manuscript and supplementary materials must be approved by the Ethics Committee. Requests can be sent to Human.Ethics.Officer@anu.edu.au.

**Funding:** CP is supported by an Australian Government Research Training Program (RTP) Scholarship. MS was funded by a research fellowship from the Westpac Scholars Trust (2019-2023). The funders had no role in the design, conduct or writing of this study.

**Competing interests:** The authors have declared that no competing interests exist.

measure and report against, performance indicators should be flexible, country-specific, and selected in consultation with immunisation actors who collect and use the data.

## Introduction

The Immunization Agenda 2030, the global strategy guiding progress on immunisation for the decade 2021–2030 endorsed by WHO member states, emphasises having "high-quality, fit-for-purpose data" to be the basis of decision-making to drive improvements in immunisation performance [1]. Greater access to data and evidence-based decision-making is considered critical to improve policies and practices, leading to better public health outcomes including higher and more equitable immunisation coverage [2].

However, the push for data-driven decision-making has led to an expanding volume of indicators for which data are collected and reported, with our recent review identifying over 600 distinct indicators being used to measure national immunisation system performance [3]. This included widely-used routine indicators on vaccination coverage (e.g. coverage of the first dose of measles vaccine) and lesser-known indicators measuring performance of various aspects of the immunisation system (e.g. proportion of districts with electronic vaccine and supply stock management systems to monitor vaccine stock). This volume of performance indicators can have unintended consequences if the resources dedicated to collecting data lead to resources being diverted away from programs [4, 5]. Furthermore, there is little evidence that these data are used to inform decision-making about immunisation programs, particularly in low-and-middle-income countries [6]. Having data for a multitude of indicators may not be useful to national-level decision-makers and program managers if the data do not inform current policy considerations, program implementation and resource allocation [2, 6, 7]. To be useful in decision-making, indicators must be aligned with the priorities of actors and the context [8], but existing monitoring and evaluation tools are rarely contextualised to a specific country or setting, raising the question of whether the top-down approach for monitoring and evaluation at a global level is appropriate [9].

With this in mind, we examined what is most useful to measure, and what is less useful or irrelevant, to assess immunisation performance in a distinct geographical region. Pacific Island Countries and Areas (PICs) comprise 21 small island countries and areas (excluding Papua New Guinea), with 3.2 million people living in an area disbursed over approximately 30% of the Earth's surface [10]. Many PICs face common challenges such as small populations (no country's population exceeds 1 million people), a highly constrained health workforce and underdeveloped information technology infrastructure [11–14]. Yet, PICs are heterogeneous, with varying population sizes (from less than 2,000 people in Niue to approximately 900,000 in Fiji), geographic size and population dispersion (e.g. concentrated in one or few islands such as Niue and Samoa, while others are dispersed over as many as 147 islands such as the Solomon Islands and French Polynesia) [10]. Local health system capacity, political and governance structures (especially as some are independent while others are affiliated with other high-income countries namely the United States, France, New Zealand and Britain), and resources also vary [14]. Most PICs used paper-based registries and tally sheets to report immunisation data prior to the COVID-19 pandemic, although some (particularly US-Affiliated Pacific Islands) had electronic immunisation registers which have enabled more varied analysis of immunisation indicators and rapid assessments of coverage among at-risk populations during outbreaks [15]. During the COVID-19 pandemic, some PICs introduced new

digital systems to collect and aggregate immunisation data pandemic, with progress in implementation varying across countries [16, 17]. While coverage of childhood immunisation was relatively high across PICs prior to the COVID-19 pandemic (average of 91.4% for the third dose of diphtheria-tetanus-pertussis (DTP) vaccine [DTP3] and 88.1% for the first dose of measles vaccine [MCV1] in 2019), noting variation across the region, many countries experienced pandemic-related disruptions to immunisation programs (coverage of DTP3 fell to 86.8% and MCV1 to 75.6% in 2021) [18, 19].

With regards to decision-making for immunisation, PICs independently evaluate and make decisions about their immunisation programs, including decisions about how immunisation services are delivered (including catch-up campaigns), new vaccine introductions, and allocation of resources. Decisions are made by staff at the country's respective ministry or department of health, with some countries having advisory and decision-making committees. No PIC has formally established a national immunisation technical advisory group due to resource constraints, and some immunisation actors contribute to decision-making both within their country and across the region. Most PICs (i.e. other than those aligned with USA or France) obtain vaccines from the Vaccine Independence Initiative with support for procurement from UNICEF [11]. Development partners (especially the World Health Organization and UNICEF with more limited support from the US Centers for Disease Prevention and Control) provide technical support in various areas including program planning, cold chain and waste management, vaccine safety and disease surveillance, both at the country-level and throughout the region. Only a handful of countries (namely Kiribati and the Solomon Islands) receive Gavi support, with financial contributions largely through vaccine procurement and technical assistance from various nongovernmental and bilateral donors. Within this complex setting, we aimed to identify which indicators are critical to measure immunisation performance in PICs and the reasons for selecting these indicators.

## Materials and methods

### Definitions

There are multiple and variable definitions for what constitutes data use, with terms often discussed interchangeably. For the purposes of this study, we used the PRISM-Act framework, and classified activities related to collecting, organising, analysing, synthesising and disseminating (including reporting) data as tasks that precede data use, collectively referred to as "data collection and reporting" [20]. "Data use" constituted actions where the data produced is reviewed, discussed and considered in planning, strategizing and decision-making, including to evaluate performance. To avoid confusion, we referred to data that is ready for use as "insights". Additionally in this paper, feasibility refers to the ease of completing the processes from data collection through to reporting, while relevance to decision-making refers to the usefulness of insights for policy and planning considerations.

### Study design

This study utilised a semi-structured expert-elicitation approach to consult with immunisation actors working in PICs and determine priorities for immunisation indicators in PICs. Expert elicitation is an approach to elicit the knowledge of experts that aims to increase accuracy and transparency in making judgements while reducing biases and quantifying uncertainty in their judgements compared with other methods of eliciting expert judgement [21, 22]. We adapted the steps detailed in the IDEA (Investigate, Discuss, Estimate, Aggregate) structured protocol for expert elicitation [21]. We adopted good practice steps identified in the elicitation literature, including: preparing the expert for the elicitation, having a formal framework for the

elicitation session, obtaining feedback from the expert on the results and offering the possibility of adjusting their response [23].

## Recruitment and study population

We invited professionals involved in delivering immunisation programs across the 21 PICs to participate in this study, including immunisation program managers, government decision-makers from ministries of health, health information specialists working with immunisation data, global development partner representatives at the regional or national levels, and international experts, advisors and researchers implementing immunisation programs or information systems in PICs. Experts were recruited between 21/11/2022 and 07/04/2023.

We purposively identified eligible individuals through publicly available information, such as lists of meeting attendees at regional immunisation meetings, as well as those known to the research team and snowballing. We contacted participants via email with a single follow-up email. We aimed to recruit 15 to 20 experts to allow for a 25% attrition rate.

## Development of elicitation instrument

The elicitation instrument comprised a list of indicators that experts were asked to rate based on specified criteria (see S1 Appendix). The list of indicators was identified from a comprehensive review of 631 distinct indicators measuring the performance of immunisation systems [3]. To make the list manageable, we only included indicators measuring outcomes of immunisation systems (focusing on full or incomplete vaccination coverage and equity of coverage, n = 35) and the performance of the systems through which those data were obtained (n = 92). We excluded indicators that were duplicated, pertained to a specific program or context, lacked clarity in their definition and calculation, and from monitoring and evaluation resources published more than 10 years prior or those preceded by another resource (n = 92). We additionally reviewed indicators categorised as measuring other components of immunisation systems but crossed over with information systems, namely regulation and pharmacovigilance (n = 3), governance and program planning (n = 2), and vaccine logistics (n = 1).

The 41 included indicators were categorised into five sections (see S2 Appendix):

1. Indicators measuring immunisation coverage (n = 12)

2. Indicators measuring insights use (n = 4)

3. Indicators measuring data quality (n = 9)

4. Indicators measuring the data systems and processes for collecting and reporting immunisation data (n = 10)

5. Indicators assessing vaccine-preventable disease (VPD) surveillance system (n = 6)

Experts were asked to rate each indicator based on whether data were feasible to collect and report, and how relevant the indicator was to decision-making. We also asked experts to select their preferred indicators within each category, and asked them to select indicators to "keep" or "remove" from a list of indicators monitoring national immunisation performance. The rating criteria are described in Table 1.

The elicitation instrument was iteratively developed and refined amongst the research team, and then revised following testing with three volunteers and piloting with three experts. Following minor revisions to improve the clarity of included questions and instructions to respondents, consistency in terminology and order of questions, the elicitation instrument was finalised.

**Table 1. Definitions of rating criteria used to elicit experts' opinions on immunisation performance monitoring indicators.**

| Rating criteria | Definition | Assessment method | Variable type | Analysis method |
|---|---|---|---|---|
| Feasibility | The ease of collecting and reporting high quality data for the indicator | For each indicator, experts were asked whether data for the indicator was: 1) Collected 2) Reported 3) Of high quality Experts could select from the following options: • Yes • No • Unsure • Not appliable | Collected as categorical; converted to numeric during analysis | For each indicator, proportions calculated for experts who stated "yes" to 1) data is collected, 2) data is reported, and 3) data is of high quality. For each individual expert, a feasibility score was calculated for each indicator by assigning a score of "1" for each "yes" response to the three items and summing them for a score out of 3. This was scaled to a score out of 10 (i.e. divide by 3, multiply by 10). A mean feasibility score for each indicator was then calculated by averaging all experts' feasibility scores for the indicator. |
| Relevance (and confidence in relevance) | Relevance: The importance of the indicator in informing decision-making about immunisation Confidence: The level of certainty in the score provided for relevance, i.e. how sure the expert felt about their relevance score | For each indicator, experts were asked to provide a numeric value between 0 and 10 (0 = lowest, 10 = highest) for both relevance and their level of confidence in that relevance score | Numeric | Crude mean relevance score for each indicator was calculated by averaging the numeric scores by each expert. Weighted relevance score for each indicator was calculated by averaging experts' adjusted relevance scores. Adjusted relevance scores were calculated for each expert by weighting their relevance score using their confidence score. |
| Preference | Indicators that experts preferred to "keep" or "remove" in a shorter list of indicators measuring national immunisation performance in PICs | Experts were asked to select a specified number of indicators (i.e. top 25% and bottom 25%, i.e. total of 11 each) to "keep" and "remove" within each of the five categories | Collected as categorical; converted to numeric during analysis | A preference score was calculated for each indicator, based on the number who selected "keep" minus the number who selected "remove". |

## Expert elicitation process

Fig 1 summarises the expert elicitation process that we followed. We conducted the elicitation over two rounds. In the first round (December 2022 to April 2023), the first author (CP) met individually with experts via web conferencing (Zoom) to discuss the purpose of the study and the criteria for rating indicators. Experts then independently completed the elicitation instrument electronically via a Qualtrics survey (see S1 Appendix). Their responses were summarised in feedback reports that were sent to experts, showing group means compared with the expert's individual responses.

In Round 2, experts participated in a qualitative discussion (via web conferencing using Zoom) to review results and explore the reasons for differences in the values given by each expert compared to the group. Discussions were completed in small groups of 2–3 experts or individually, based on experts' availability and preferences (individual results were anonymised). Discussions were structured to review indicator rankings within each of the five categories, and then the overall highest ranked indicators. Open-ended questions were framed to understand the reasons behind experts' ratings, the context they worked in, their professional experiences and the assumptions being made. Specific questions included why they thought an indicator was ranked high (or low), and how they might use an indicator in their context. Experts were also asked about their preferences for characteristics of sets of indicators, specifically how many indicators they think a national set of immunisation performance indicators should be limited to and why. Every expert was then sent a brief summary of all discussions, and asked to rate all 41 indicators a second time (June to August 2023). The entire study, including instrument administration and discussions, was conducted in English only.

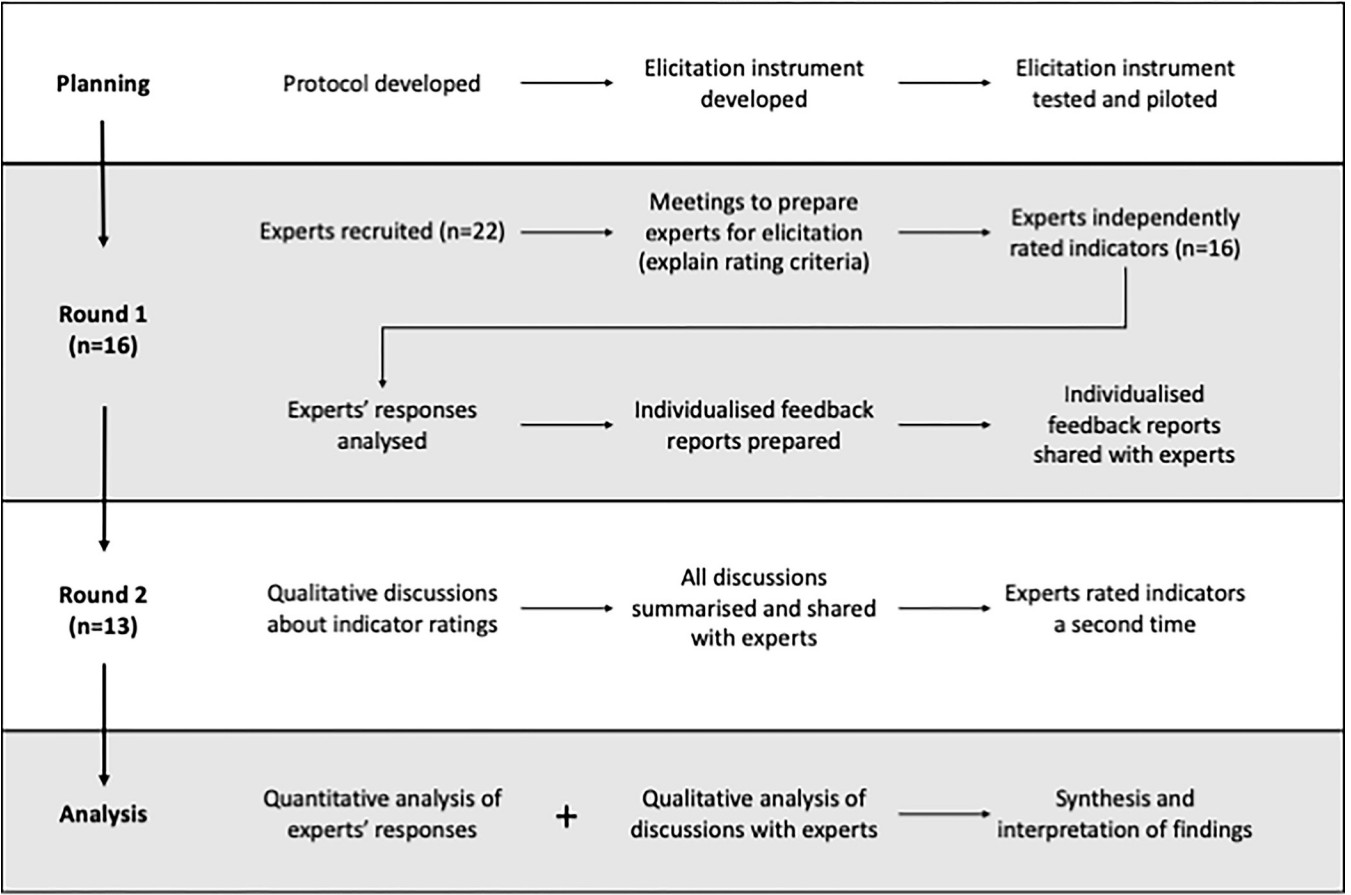

**Fig 1. Flowchart showing the process of expert elicitation followed in this study.**

## Ethical approval

The Australian National University Human Research Ethics Committee reviewed and approved the ethical conduct of this study (protocol 2022/368) and amendments to data collection tools. Informed consent was obtained verbally during the first meeting. Experts were asked to confirm their consent to participate in writing at the time they completed the elicitation instrument online.

## Analysis

We included results for experts who completed both rounds of the elicitation only. We analysed indicator ratings descriptively, calculating the proportion of experts who stated that data could be collected, reported and were high quality for each indicator. We generated a feasibility score for each expert's indicator rating, by assigning a score of 1 for each "yes" response to the three feasibility items and summing these for a score out of 3, and then scaling this to a total out of 10. We generated mean relevance scores for each indicator by weighting the crude mean relevance scores assigned by experts with their confidence scores. We then generated a composite feasibility-relevance score for each indicator by averaging the mean feasibility and relevance scores.

We calculated a "preference score" for each indicator by subtracting the number of experts who said they would "remove" the indicator from those who said they would "keep" that

**Table 2. Characteristics of included study participants (n = 13).**

| Characteristic | n | % |
|---|---|---|
| Gender of respondent | | |
| Male | 6 | 46.2 |
| Female | 7 | 53.9 |
| Age of respondent | | |
| 18–29 years | 2 | 15.4 |
| 30–39 years | 4 | 30.8 |
| 40–49 years | 4 | 30.8 |
| 50–59 years | 2 | 15.4 |
| 60 years or older | 1 | 7.7 |
| Time working in PICs | | |
| <1 year | 1 | 7.7 |
| 1–<2 years | 1 | 7.7 |
| 2–<5 years | 4 | 30.8 |
| 5–<9 years | 3 | 23.1 |
| ≥10 years | 4 | 30.8 |
| Type of expert | | |
| Ministry/department of health employee | 5 | 38.5 |
| Global development partner representative | 2 | 15.4 |
| Consultant or researcher | 6 | 46.2 |
| Subject matter focus area | | |
| Immunisation and public health | 10 | 76.9 |
| Digital health | 3 | 23.1 |
| Geographical area of focus | | |
| Global or regional | 7 | 53.8 |
| National | 6 | 46.2 |
| Time working in current role | | |
| <1 year | 0 | 0.0 |
| 1–<2 years | 6 | 46.2 |
| 2–<5 years | 2 | 15.4 |
| 5–<9 years | 4 | 30.8 |
| ≥10 years | 1 | 7.7 |

indicator. We then used the preference scores and composite feasibility-relevance scores to rank the 41 indicators. Table 1 and S3 Appendix provides details on how these scores and rankings were calculated. We conducted a sensitivity analysis to examine the effect of using different rating criteria to rank indicators. Analyses were conducted in Stata 17 [24], and graphics were produced using R [25].

We additionally examined indicator rankings by different expert groups, namely by gender, expertise (public health and immunisation versus digital health) and geographical scope of work (one country versus more than one country), and describe findings narratively. We examined the relationships between the different rating criteria using graphs and calculated correlation coefficients between the rating criteria.

To understand the reasons driving preferences for indicators, we analysed the discussions with experts from Round 2 using framework analysis, following the steps outlined by Ward et al. [26]. We used an iterative process and inductive approach to identify themes. One author (CP) reviewed the interview transcripts and summarised salient discussion points, which were used to identify an initial set of themes, organised into overarching concepts (constructs)

which were discussed and refined by all authors. Two authors (CP and AT) reviewed three transcripts (covering 7/13 participants) in detail to test how well the themes fit. The themes were revised following discussion with all authors. CP and AT then independently coded three interview transcripts (covering 5/13 participants) and discussed discrepancies in coding. Interviews were coded using NVivo 12. After further discussion with authors, the themes were refined and finalised. A single author (CP) then coded remaining interviews based on the agreed themes and definitions (see S4 Appendix), and extracted data into a framework examining themes by each expert and their context. Findings are summarised narratively.

## Results

Twenty-one experts agreed to participate in the study– 16 (72.7%) completed the first round of which 13 (81.3%) completed the second round and were included in the final analysis. Table 2 shows the characteristics of included experts. Seven experts worked in multiple Pacific Island countries or across the region, while six worked in a single country. Specific countries represented included Fiji, French Polynesia, Kiribati, Niue, Samoa, Solomon Islands and Vanuatu. Ten were experts in immunisation and public health, and three were experts in health information systems. Most (11/13, 85%) had worked in Pacific Island countries for more than 2 years. Among those who initiated (i.e. met with the research team and provided verbal agreement to participate or completed part of the study) but did not complete their participation in the study, five were affiliated with development partner organisations or research institutions, while three were country-level staff with their respective ministries of health (one immunisation expert and two health information specialists).

### Results of indicator ratings and rankings

Table 3 shows the summary of preference, feasibility and relevance scores for each indicator. Feasibility scores ranged between 0.77 to 7.18 (details in S5 and S6 Appendices), with only 9 indicators receiving a score above 5. Mean weighted relevance scores varied between 4.97 and 9.05, noting that most indicators (33/41, 80.5%) received mean crude and weighted scores between 7 and 9 (see S5 Appendix). Preference scores were moderately correlated with composite feasibility-relevance ($r = 0.68$), weighted relevance ($r = 0.68$) and feasibility ($r = 0.56$) scores (Fig 2). The two attributes examined in this study, feasibility and relevance to decision-making, were moderately correlated ($r = 0.52$, Fig 3).

The ten highest-rated indicators included indicators from all five categories. The highest-ranked indicator, "Country uses quality data on under-vaccinated to inform plans at community, subnational and national levels", received the highest mean relevance score (9.05/10). Among immunisation coverage indicators, those measuring measles vaccine coverage and zero-dose vaccination coverage were the most highly rated (mean relevance–MCV1 coverage: 8.29, MCV2 coverage: 8.98, zero-dose coverage: 8.47). Indicators examining the establishment of systems to monitor immunisation coverage (mean relevance: 9.01) and adverse events following immunisations (mean relevance: 8.97) were amongst the highest-rated of indicators measuring the performance of data systems for immunisation. Among the fourteen indicators ranked the highest overall (i.e. top one-third of all 41 indicators), all except two ('proportion of facility-level routine immunisation microplans with updated catchment area maps and strategy to reach them' and 'linkage of home-based records with civil birth registration through immunisation services') had relevance and feasibility scores above the median scores (see Fig 2). In our sensitivity analysis, the highest ranked indicators remained largely unchanged, with the most differences noted when indicators were ranked by feasibility scores first (see S7 Appendix).

**Table 3. Indicators listed by their rank based on scores provided by experts for preference, mean feasibility and mean relevance-to-decision-making.**

| Rank | Indicator | Category | Preference score | Mean feasibility-relevance score | Mean feasibility score | Mean weighted relevance score |
|---|---|---|---|---|---|---|
| 1 | Country uses quality data on under-vaccinated to inform plans at community, subnational and national levels | Use of insights | 10 | 6.96 | 4.87 | 9.05 |
| 2 | Proportion of live births registered | Data quality | 6 | 8.05 | 7.18 | 8.92 |
| 3 | Number of districts with measles (MCV1) coverage in each range: <50%, 50–79%, 80–89%, 90–94, ≥95% | Immunisation coverage | 6 | 7.35 | 6.41 | 8.29 |
| 4 | Number of districts with measles (MCV2) coverage in each range: <50%, 50–79%, 80–89%, 90–94, ≥95% | Immunisation coverage | 6 | 7.18 | 5.38 | 8.98 |
| 5 | Is there a national system to monitor adverse events following immunisation (AEFIs)? | Data systems and processes | 6 | 6.79 | 4.62 | 8.97 |
| 6 | Availability of sustainable and effective immunisation information system integrated within a robust national health information system | Data systems and processes | 6 | 6.43 | 3.85 | 9.01 |
| 7 | Number of zero dose children, i.e. those that lack access to or are never reached by routine immunisation services (operationally measured as those who lack first dose of a DTP-containing vaccine) | Immunisation coverage | 5 | 7.31 | 6.15 | 8.47 |
| 8 | Proportion of polio, measles, meningococcal disease, yellow fever, cholera, and Ebola outbreaks with timely detection and response | VPD surveillance systems | 4 | 6.34 | 3.85 | 8.84 |
| 9 | Proportion of facility-level routine immunisation microplans with updated catchment area maps and strategy to reach them | Data quality | 4 | 4.69 | 2.05 | 7.34 |
| 10 | Linkage of home-based records with civil birth registration through immunisation services | Data systems and processes | 3 | 5.01 | 2.56 | 7.45 |
| 11 | Dropout rates between first dose (DTP1) and third dose (DPT3) of DTP-containing vaccine | Immunisation coverage | 2 | 7.13 | 5.90 | 8.36 |
| 12 | Proportion of districts with complete and timely reporting from all health facilities | Data quality | 2 | 6.93 | 6.15 | 7.70 |
| 13 | Non-polio acute flaccid paralysis (AFP) rate (target >1/100,000 among <15 years population) in a 12-month period | VPD surveillance systems | 2 | 6.36 | 4.36 | 8.36 |
| 14 | Proportion of districts having electronic vaccine and supply stock management system to monitor vaccine stock down to service delivery | Data systems and processes | 2 | 5.45 | 2.05 | 8.84 |
| 15 | Proportion of districts reporting negative DTP1-DTP3 drop out | Data quality | 2 | 5.20 | 3.59 | 6.80 |
| 16 | Dropout rates between first dose (DTP1) and first dose of measles-containing vaccine (MCV1) | Immunisation coverage | 1 | 6.81 | 4.87 | 8.75 |
| 17 | Access to laboratory capacity to test for at least one bacterial vaccine-preventable disease (VPD) | VPD surveillance systems | 1 | 6.02 | 3.85 | 8.19 |
| 18 | Proportion of children with home-based immunisation records | Data systems and processes | 1 | 4.72 | 2.31 | 7.14 |
| 19 | Number of districts with DTP3 coverage in each range: <50%, 50–79%, 80–89%, 90–94, ≥95% | Immunisation coverage | 0 | 6.96 | 5.90 | 8.03 |
| 20 | Proportion of districts reporting stock availability (vaccines and supplies) at a service delivery level | Data systems and processes | 0 | 6.58 | 4.36 | 8.80 |
| 21 | Proportion of stockpile applications that demonstrate use of evidence (e.g. disease surveillance data, root cause analysis, and coverage data) to support planning/targeting of outbreak response campaigns | Use of insights | 0 | 6.32 | 4.36 | 8.28 |
| 22 | Number of districts reporting DTP drop out ranges greater than 10%, by coverage range: <50%, 50–79%, 80–89%, 90–94, ≥95% | Immunisation coverage | 0 | 6.29 | 4.62 | 7.96 |
| 23 | Proportion of districts reporting at least 90% on time during a one-year period for suspected cases for all priority vaccine-preventable diseases under nationwide surveillance, including reporting of zero cases | Data quality | 0 | 6.24 | 4.36 | 8.12 |

*(Continued)*

**Table 3.** (Continued)

| Rank | Indicator | Category | Preference score | Mean feasibility-relevance score | Mean feasibility score | Mean weighted relevance score |
|---|---|---|---|---|---|---|
| 24 | Does the country collect age and/or number of vaccine doses received for all cases of vaccine-preventable disease? | VPD surveillance systems | 0 | 5.58 | 3.08 | 8.09 |
| 25 | Proportion of districts with complete and timely reporting | Data quality | -1 | 6.72 | 5.64 | 7.79 |
| 26 | Are the number of type-specific vaccine doses reported by age group (e.g. number of diphtheria cases by age group) based on recall, documentation, or both? | Data quality | -1 | 6.68 | 5.13 | 8.24 |
| 27 | Annual number of laboratory-confirmed epidemic-prone vaccine-preventable disease outbreaks | VPD surveillance systems | -1 | 6.27 | 3.85 | 8.70 |
| 28 | Number of districts with protection at birth (PAB) (against neonatal tetanus) coverage in each range: <50%, 50–79%, 80–89%, 90–94, ≥95% | Immunisation coverage | -1 | 5.83 | 4.36 | 7.29 |
| 29 | Proportion of districts with on-line access to health management information systems (HMIS) | Data systems and processes | -3 | 5.68 | 2.82 | 8.54 |
| 30 | Proportion of eligible children in the disadvantaged population that are reached and vaccinated according to national schedule | Immunisation coverage | -3 | 4.31 | 1.28 | 7.33 |
| 31 | Proportion of district health management committees (or equivalent at subnational level) that review immunisation performance as part of primary health care performance at least annually | Use of insights | -4 | 4.28 | 1.79 | 6.76 |
| 32 | DTP3, MCV1, and MCV2 coverage in the 20% of districts with lowest coverage | Immunisation coverage | -5 | 6.11 | 4.62 | 7.61 |
| 33 | Number of districts reporting DTP drop out ranges greater than 10% | Immunisation coverage | -5 | 5.82 | 3.85 | 7.78 |
| 34 | Individual adverse event following immunisation (AEFI) case safety reports per million total population | Data systems and processes | -5 | 5.40 | 3.33 | 7.47 |
| 35 | Proportion of population with access to their personal immunisation records | Data systems and processes | -5 | 4.01 | 0.77 | 7.26 |
| 36 | Non-measles/non-rubella discard rate (target ≥2/100,000 population) | VPD surveillance systems | -6 | 5.81 | 3.85 | 7.77 |
| 37 | Percentage points difference in coverage of DTP1, MCV1 and full immunisation coverage associated with the most important socioeconomic determinants of vaccination coverage in the country (poverty, education, ethnicity, religious affiliation) | Immunisation coverage | -6 | 4.53 | 2.05 | 7.00 |
| 38 | Does the private health sector deliver vaccines in your country and do you report it in your coverage? (Private health sector includes all organisations not owned or controlled by governments, including for-profit or not-for-profit, formal or informal, and domestic or foreign.) | Data quality | -6 | 4.28 | 3.59 | 4.97 |
| 39 | Proportion of districts with year-to-year variation of children vaccinated with DTP3 less than 15% | Data quality | -6 | 3.96 | 1.79 | 6.12 |
| 40 | Commitment tracking and accountability frameworks used at country and subnational levels | Use of insights | -6 | 3.94 | 0.77 | 7.11 |
| 41 | Proportion of provinces/districts or other subnational units with at least one documented (with reporting form and/or line listed) individual serious AEFI case safety reports per million total population | Data systems and processes | -6 | 3.86 | 1.03 | 6.69 |

The highest-ranked indicators were largely similar between genders and professional groups (i.e. professional expertise and geographical scope of work, see S8 and S9 Appendices). Zero-dose coverage was the second-highest ranked indicator among experts working in more than one PIC, but ranked lower by nationally-based experts and health information specialists (ranked 14th and 17th, respectively). Monitoring the establishment of adverse events

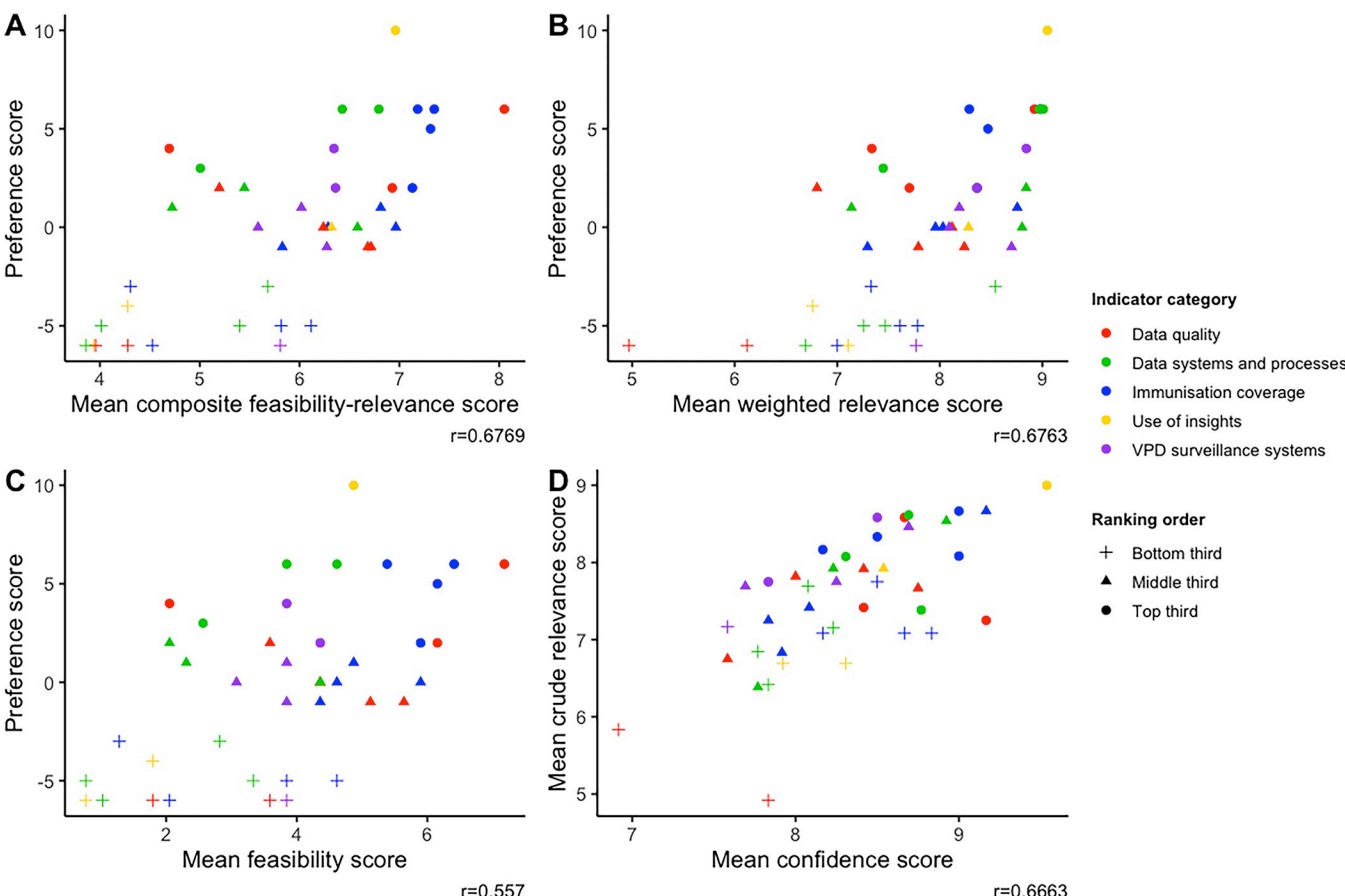

**Fig 2. Plots comparing mean scores by rating criteria for all indicators.** The graphs show how preference scores varied by the rating criteria used in this study (i.e. feasibility, relevance, and a composite feasibility-relevance score) for each indicator. They show moderate correlations of preference scores with each of the rating criteria. Notes: Top third: Highest ranked indicators (top one-third, indicators that ranked 1–14). Middle third: Middle-ranked indicators (middle one-third, indicators that ranked 15–29). Bottom third: Lowest-ranked indicators (lowest one-third, indicators that ranked 30–41). r = correlation coefficient between rating attributes. A. Preference scores vs composite feasibility-relevance scores. B. Preference score vs relevance score (weighted). C. Preference score vs feasibility score. D. Crude relevance vs confidence.

surveillance was the highest ranked indicator among nationally-based experts, but ranked 20th among experts working in more than one PIC. Both groups ranked the use of data on under-vaccinated to inform planning and measles vaccination coverage amongst the top ten of all 41 indicators. Health information specialists ranked indicators related to the timeliness and completeness of immunisation data higher than immunisation specialists (5th and 6th versus 15th and 27th), albeit noting the small number of health information specialists in the study limits interpretation of this finding.

### Factors influencing preferences for indicators

Experts highlighted two major reasons for preferencing indicators: 1) usefulness for decision-making and planning, and 2) ease of data collection, reporting and interpretation.

**1. Usefulness for decision-making and planning.** During the discussions, experts repeatedly expressed that they selected indicators that inform policy considerations to strengthen immunisation programs and make programmatic decisions. Experts discussed how they used certain insights in decision-making to develop microplans and target resources, stating that

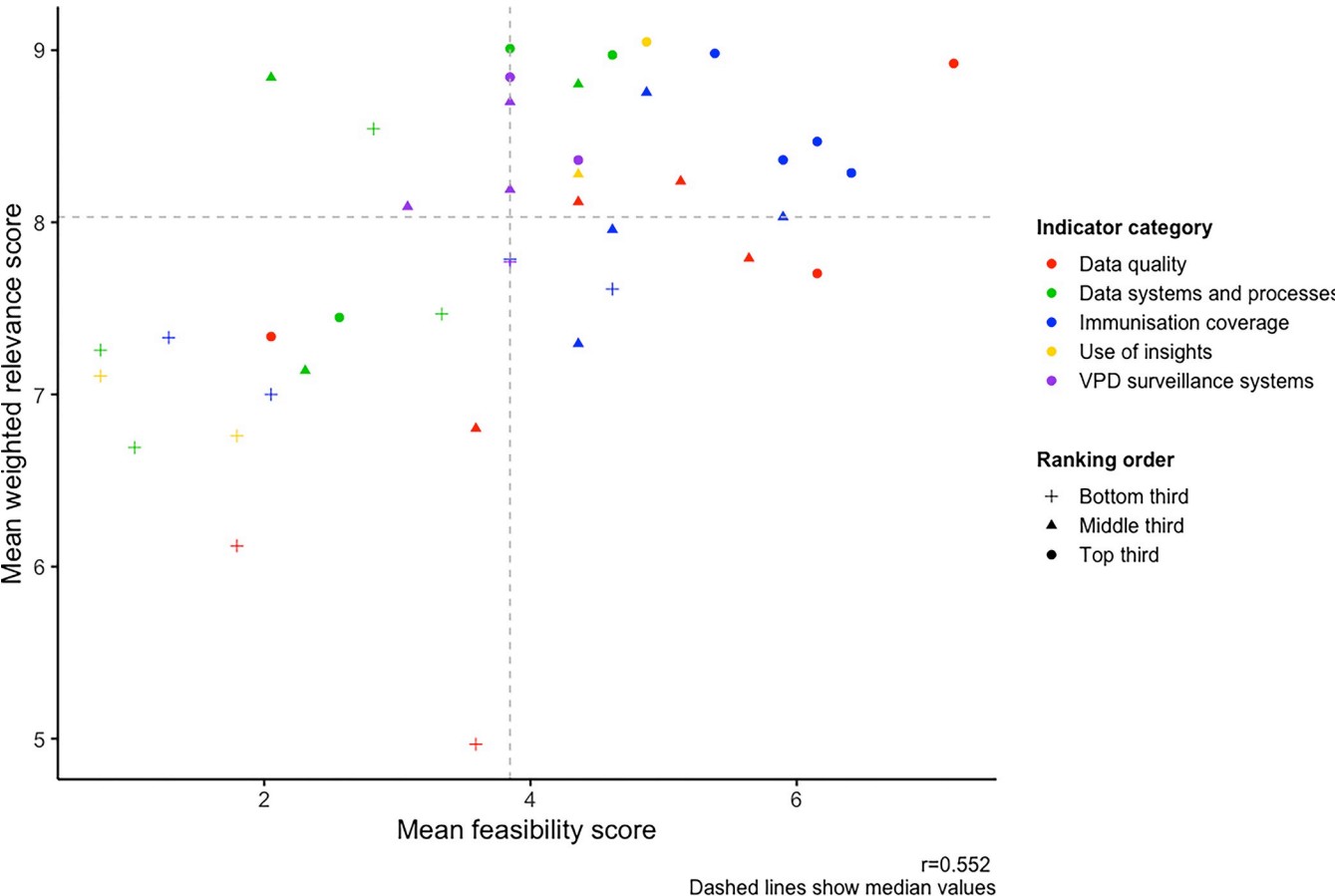

**Fig 3. Plot showing mean weighted relevance scores and mean feasibility scores for all indicators.** The graphs show how scores for the two rating criteria used in this study, i.e. feasibility and relevance (weighted by confidence scores), varied for each indicator. The graphs show a moderate correlation between mean feasibility and mean weighted preference scores. Notes: Top third: Highest ranked indicators (top one-third, indicators that ranked 1–14). Middle third: Middle-ranked indicators (middle one-third, indicators that ranked 15–29). Bottom third: Lowest-ranked indicators (lowest one-third, indicators that ranked 30–41). r = correlation coefficient between rating attributes.

their preferred indicators helped to devise strategies to reach unvaccinated children and plan for delivery including managing logistics and supply chains. Experts wanted data that led to action and helped inform strategies to improve immunisation:

> "I mean, it's talking about using data for action, right? So that all the data that we are collecting, I think countries are interested in, how does it contribute to policy? How does it contribute to program improvement? How does it contribute to reaching children and communities with vaccines? So that's very interesting about it. And I think that this, that shows you that all stakeholders are interested in the data not just being collected, but actually being used to inform planning." (Immunisation expert working across multiple PICs)

Experts involved in planning immunisation programs highlighted the importance of having data on stock availability and management at all levels of the health system to ensure immunisation services aren't interrupted, to take action to understand the cause of stockouts and resolve them promptly, and to undertake planning to minimise stockouts in the future. Additionally, experts highlighted the need to know where populations, particularly

undervaccinated children, are to develop microplans for service delivery and target resources appropriately:

*"This one was important, more so than the other indicators, because it shows you know where the unvaccinated children are, and what we need to do to go and reach them, what resources you need, how many staff you need, how much diesel you need, how many boats you need to actually access those communities." (Immunisation expert working across multiple PICS)*

Experts frequently discussed that their indicator preferences were driven by a desire to know where performance was suboptimal as this helped to keep the issue on the agenda and ensure resources are allocated to improve performance. For example, two experts discussed how in their respective countries, coverage of vaccines doses in multi-dose schedules (i.e. DTP and measles vaccines) was high and stable in the first year of life, but there was high dropout and coverage of doses given later in life was lower. Having insights on the coverage of these later doses was valued more as it helped to identify where problems were specifically occurring (e.g. which provinces) and make decisions about where to target resources, as well as advocate for additional resources from higher-level decision-makers and politicians. Other experts discussed that knowing coverage rates for the first dose of measles (or valuing both first and second doses equally) was their preference because their primary concern was preventing measles outbreaks. Experts involved in planning programs indicated how low measles coverage was a trigger for them to decide whether supplementary immunisation activities were required.

In an example of system-level initiatives, an expert cited how the proportions of births registered in their country was found to be low and that this insight helped to drive actions to improve birth registrations. This expert valued continued monitoring of that indicator to ensure that there was continued progress, and prompt action in case performance worsened. In another example, experts discussed how reporting of adverse events following immunisation was known to be problematic in their setting, and so they prioritised indicators related to examining if the safety surveillance system was established and if events were being reported.

Experts highlighted additional areas where challenges are known that should be monitored as they were integral to immunisation system performance. Limited health worker capacity was frequently noted, with experts discussing how shortages in nursing staff hindered the ability to administer vaccinations. High turnover resulted in the program being implemented by inexperienced staff who lack competencies in using insights effectively to identify unvaccinated children. Vaccine shortages and stockouts were also noted to disrupted vaccination efforts, with monitoring necessary to determine the origin of the problem (e.g. whether the stockout was due to poor stock management at the facility level or a national issue due to lack of payment to the manufacturer) and therefore the solution.

Experts discussed their preference for indicators that measure the performance of areas where there have been recent efforts to improve systems, particularly establishing robust surveillance systems for adverse events following immunisations and digital individualised immunisation information systems. Experts explained that the need for these systems during the COVID-19 pandemic had attracted substantial investment and political will to establish them, so these were priority areas where countries wished to demonstrate progress and achieve targets. Experts also explained that establishment of these systems required multiple elements of the health system to come together, so achieving these goals demonstrated health system strengthening.

**2. Ease of data collection, reporting and interpretation.** Experts expressed a strong preference for indicators for which data are relatively easier to collect and analyse, and identified

difficulties in collecting, reporting and interpreting data as reasons not to select indicators. Factors that affected the feasibility of data collection and reporting included whether the data were currently captured by existing health information systems, difficulties in defining sub-population groups, having sufficient resources for data collection and reporting, lacking data systems that enable easier reporting, and complexity of synthesis. Workforce constraints were frequently cited as a barrier, with experts noting the need to be pragmatic about what insights can be generated as the resources required were either absent or incommensurate with the benefits of having those insights. This was especially highlighted in discussions regarding reporting coverage data by geographical and socioeconomic groups, as collecting and report-ing data at more granular levels was considered particularly resource-intensive:

> *"This may be a distinction that needs to be made between whether or not, the indicator is important, and like, how difficult it would be to get the information, accurate information to, to report against that indicator. I think in my case, I probably just wanted to remove the indi-cators that I felt, which is beyond the scope of the capacity, the organisation to be able to go out and collect rather than necessarily being important. So I think it's like the socioeconomic determinants, vaccination coverage might be really providing very valuable information. But to get to that information is just very complicated and challenging." (Health information spe-cialist working in a single PIC)*

> *"Regarding managers and people dealing with data, we are very few. For example, in [PIC name omitted], for [hundreds of thousands] people, I am alone to manage all that. And not only the vaccines, but also other ID [infectious disease] programs." (Immunisation manager working in a single PIC)*

Experts also voiced a preference for indicators that were simple and straightforward to interpret. Experts objected to indicators that were poorly defined, lengthy, had complex word-ing, had multiple layers or could be interpreted in multiple ways (such as "Percentage points difference in coverage of DTP1, MCV1 and full immunisation coverage associated with the most important socioeconomic determinants of vaccination coverage in the country"). Com-plex, wordy indicators were considered to be confusing, ambiguous and lacking clarity in what they were measuring, which presented challenges both in using insights but also in determin-ing how to collect the data. Terms such as "disadvantaged population", for example, were not clearly defined and interpreted differently by experts, with some stating that everyone in their country had access to immunisations so there was no "disadvantage" as such.

## Context-specific factors influencing indicator preferences

Factors that explained differences in experts' preferences for indicators fell into four themes: A) differences in health priorities, B) differences in health system structures, C) priorities and experiences of different immunisation actors, and D) challenges due to small and mobile populations.

**A. Differences in health priorities.** Experts differed in how they rated certain indicators as they either did not align with what they believed were urgent priorities for immunisation or because they did not think it would help them improve immunisation coverage. One such indicator was the number of zero-dose children (defined as those lacking a dose of DTP-con-taining vaccine), with several experts considering it important to report data on this indicator to ensure no child is unvaccinated. However, perspectives on the extent of zero-dose vaccina-tion as a public health problem in specific PICs varied, and was believed to be more important in countries with lower coverage of routine childhood vaccinations and less so for those with

high coverage. Some felt that there were very small numbers of unvaccinated children in their country so the relative benefit of having this indicator and spending resources on finding and vaccinating these children was small. Others felt that zero-dose children were more common than statistics indicated, with coverage rates believed to be inaccurately inflated due to issues with population denominators. One expert suggested that simply knowing there were unvaccinated children was insufficient, and that the reasons why they were un- or under-vaccinated was just as important. Another expert raised that closely tracking coverage of booster doses of DTP vaccine would be more influential in strengthening their program, as coverage of the fourth, fifth and sixth doses of DTP vaccine were known to be low whereas coverage of DTP vaccine doses given in infancy is high. Some also expressed concern about focusing on coverage of one type of vaccine, when full vaccination was the goal.

Another indicator where preferences varied was for the proportion of live births registered. Some experts viewed this as a priority for improvement as it influenced the calculations for all coverage and other health indicators, while others did not see how this insight could help improve immunisation performance and believed other sources of birth data could be used for follow-up of children due for vaccination. Similarly, indicators measuring VPD surveillance capacity were viewed by some experts to be important for VPD detection and outbreak management, but viewed by others as beyond the scope of the immunisation program and therefore irrelevant for monitoring immunisation performance.

**B. Differences in health system structures.** Experts discussed how differences in health system structures and maturity affected preferences for indicators, particularly those measuring the performance of data systems. Some noted that French-overseas collective PICs and US-Affiliated Pacific Islands have additional resources enabling them to have more advanced systems with digital capabilities. Other countries are in earlier stages of establishing new systems for immunisation data and only just starting to digitise information systems, so measuring data quality or enabling linkage of different data systems are secondary to ensuring that effective systems to record data are established. Some experts discussed how capabilities vary even within countries, with the use of digital technologies to record data limited by infrastructural issues like electricity and internet connectivity.

Another example where health system factors affected indicator preferences was for indicators measuring VPD surveillance systems. Experts agreed that laboratory capacity to detect VPDs is a gap across the Pacific Islands, but differed on how important they considered this to be. Some thought it is a major problem for detecting and responding to outbreaks, often receiving the diagnosis for a suspected measles case several weeks after the first case. Others acknowledged that increasing laboratory capacity in some countries, especially the smallest PICs, was unlikely to be practical or cost-effective, thus indicators measuring laboratory capacity were irrelevant in those contexts.

**C. Priorities and experiences of different immunisation actors.** Experts discussed differing preferences based on their roles, experiences and sphere of influence. Some experts, including those from global development organisations, noted the tension created by the requirements for reporting imposed by external actors, noting there were often discrepancies in what was interesting to donors and funders, but not necessarily feasible and meaningful to collect from countries' perspectives. Experts expressed frustration that development partners did not always agree on the number of indicators or the types of indicators that are important, creating additional work and confusion.

Among experts working in a single PIC, preferences for indicators were affected by their scope of work and level of responsibility. This was particularly discussed in relation to the indicator about the completeness of birth registration data–while some experts considered this important to be able to interpret immunisation and other health statistics, others considered

this beyond their scope of work in immunisation and not wanting to be accountable for a performance measure they could not influence.

**D. Challenges due to small and highly mobile populations.**   Experts repeatedly cited the small population sizes in PICs as being the reason why some indicators were irrelevant or difficult to interpret. With small population sizes and therefore a smaller workforce with a few people doing the same work of multiple people in larger countries, certain governance structures used in larger countries were considered redundant or unnecessary. For example, experts noted that subnational-level governance committees were impractical, and in fact even at the national level there were insufficient skilled workforce to establish national immunisation technical advisory groups in each PIC. Small population sizes also made interpretation of some statistics challenging, with experts noting that denominators of "per million" (e.g. AEFI reports per million) were especially troublesome given that PICs have populations much less than one million people. This was particularly the case for rare events, where a single additional serious AEFI case can mean a large rise in rates.

Experts especially cited difficulties in generating data by granular geographical areas as problematic. None of the experts supported reporting statistics by district, citing its irrelevance to decision-making in their respective countries, and difficulties in collecting, reporting and interpreting data at this level. They explained that scenarios such as a child being born in one district (where their birth is registered), living in a second, and going to school in a third were common, with vaccinations administered being captured in an area other than the child's birth district. Thus, low coverage rates, rates over 100% and negative dropout rates at subnational levels reflect movements rather than poor performance or inaccuracies. Some experts stated that in their context, other subnational denominations, like province, island or health facility catchment area, were better aligned with the way health service delivery was organised and therefore more useful for planning services and identifying pockets of under-vaccinated populations. For higher level decision-making, experts preferred to routinely monitor insights at the national level, while conducting deep dives into specific areas if a problem is detected or suspected. Experts also noted that subnational denominations were irrelevant in PICs where the population sizes are very small and health services are highly centralised.

## Characteristics of a set of indicators

As the original objective of this study was to create a single set of indicators suitable for PICs, we asked experts how many indicators they believe a single set of indicators to monitor immunisation performance nationally should be limited to (see Box 1). Experts unanimously preferred to have smaller sets of indicators, citing the burden of data collection and reporting in a small island nation context. Experts considered between ten and 15 indicators to be sufficient and manageable for national immunisation performance monitoring, with more considered tedious and challenging to report. Less data were also considered to be more influential, with only limited insights being used to demonstrate a need or performance improvements, or to make decisions. Experts said fewer insights to be more likely to be reviewed and used by decision-makers in contrast to data on a large volume of indicators, and therefore lead to concrete actions. Some experts stated that having too many insights could inadvertently lead to an aversion to using them. Experts also preferred including indicators that encompassed a broad range of outcomes rather than having indicators that provided depth of information on a specific outcome. They viewed in-depth investigations as being a second step and more appropriate if suboptimal performance was detected at the national level, particularly given the small populations in PICs and practical difficulties of obtaining insights at subnational levels.

Box 1. Desirable characteristics of a set of indicators monitoring national immunisation performance in Pacific Island Countries and Areas as identified by experts in this study

Indicator characteristics:

- Relevance to decision making: informs program planning, demonstrates progress (or lack thereof) in an area of focus or importance, identifies problems, informs a public health action

- Relates to a priority area

- Feasible to collect, analyse, and report within the resources and health and information technology infrastructure available

- Simply worded and easy to understand and interpret

- Meaningful in the context where it is being used

- Reported data are of high quality (i.e. accurate, reliable, trustworthy)

Characteristics of sets of indicators:

- Provides a broad (but not necessarily in-depth) overview of immunisation performance

- Fewer indicators–between 10 to 15 indicators (up to 20, depending on the context)

Included indicators are not redundant

## Discussion

Our study found substantial variation in which indicators experts believed were most important for immunisation performance in the Pacific Island region. This was contrary to our initial hypothesis that a common set of indicators could be formed for this geographical region with many shared constraints. Despite this, we found that experts had similar reasons for selecting indicators, with differences in indicator preferences reflecting differing immunisation goals and health system constraints across countries. We found relevance to decision-making correlated with preferences for indicators, consistent with findings from other studies where experts have selected indicators to measure healthcare performance [8, 27–30]. Experts in our study considered indicators to be more relevant if they monitored areas of priority and if they led to public health action. Feasibility was also an important consideration that experts cited as a factor to screen indicators in or out. The burden of data collection and reporting is well-documented in resource-constrained countries [2, 7, 31, 32]. For Pacific Island countries with small, dispersed populations and a constrained workforce hampered further by skilled personnel emigration [33, 34], the opportunity costs of collecting and reporting immunisation data are even higher.

There were certain indicators, or areas of measurement, that experts broadly agreed were relevant and important to measure across the Pacific. Indicators for measles vaccination coverage, for example, were unanimously agreed to be the most important coverage indicator, influenced by the recent measles epidemic in the Pacific region [35, 36] and concerns about outbreaks following disruptions to routine immunisation during the COVID-19 pandemic

[37–39]. There was also agreement that some indicators were not useful to decision-makers, such as coverage indicators at specific subnational divisions.

Experts differed in their preferences for indicators when the insight lacked utility in a specific context, and were not always in favour of some indicators adopted by the global immunisation community to strengthen immunisation. For example, zero-dose vaccination coverage has been promoted as a means to improve equity and universal health coverage [40, 41], but not all experts perceived reaching unvaccinated children to be a challenge in their context and therefore less valuable in their setting. While zero-dose coverage is operationally defined as those lacking a first dose of DTP vaccine for simplicity, questions have been raised about the best way to define "zero-dose" both by experts in our study and more broadly [42]. Similarly, experts agreed that preventing measles and strong surveillance capabilities were a high priority, but cited challenges with increasing laboratory capacity for testing that rendered indicators like the non-measles discard rate less useful. The lack of feasibility of increasing laboratory capacity to global standards has been a challenge to verification of measles elimination in PICs [43], highlighting how targets reasonable for larger countries may be unattainable, and possibly unnecessary, for PICs.

Standardising data reporting at the global level has the benefit of being able to compare performance across countries' and time, encouraging visibility and accountability to achieve progress towards immunisation goals [44]. This is particularly true for targets that are well-established markers of immunisation performance like DTP vaccine and MCV coverage [45]. Standardising immunisation performance indicators globally also has the advantage of providing a common framework, making monitoring and evaluation mainstream practice, and drawing attention to issues with data quality [44]. However, a one-size-fits-all approach to immunisation performance monitoring wastes finite health resources, and may conversely have the adverse impact of discouraging the use of insights especially at subnational levels [46]. Frustration about reporting data to external organisations when the insights were not useful to decision-makers was unanimous among our experts. Externally-imposed demands for data limits resources to measure indicators that are more informative to decision-makers, a problem that has been reported even among large healthcare organisations in high-resource settings [5].

While the Immunization Agenda 2030 Monitoring and Evaluation Framework (which guides global immunisation monitoring) allows for some flexibility in selecting indicators at the national level, in practice countries are required to submit data on a vast number of indicators. To minimise the burden of data collection, one approach to global immunisation performance measurement could be to mandate reporting on a minimum set of core indicators, supplemented by optional indicators that countries can select to suit their country context. Core indicators should have clear evidence demonstrating that their achievement leads to public health benefits like improvements in vaccination coverage, reductions in disease morbidity and mortality, or the advancement of universal health care. An alternative approach is to mandate measuring performance in a specific area (e.g. vaccination coverage or AEFI surveillance) without mandating reporting of a specific indicator.

## Strengths and limitations of the study

We used an expert elicitation methodology for our study as it allowed us to use a systematic approach to identifying indicator preferences while quantifying the degree of uncertainty in indicator ratings, acknowledging the differing roles and level of experience of participants. The qualitative component allowed us to identify reasons to explain the lack of consensus on which indicators are preferred, and helped to identify a common set of criteria that experts use to determine which indicators are best used to measure immunisation performance.

We limited the criteria used to rate indicators to feasibility and relevance to decision-making, which we considered to be the most important criteria to identify context-specific indicators and are most frequently used in similar studies [47, 48]. Other studies include additional criteria, such as validity, reliability and significance/importance [8, 28–30, 48]. We did not include attributes of the indicator itself as criteria as we identified indicators in existing monitoring and evaluation resources, and considered that they had been through an appraisal process. Preferences for indicators are complex and may be affected by a variety of factors that may not be possible to capture through specific criteria. One of the strengths of our method is that we asked experts, all things considered, which indicators would they prefer. This aimed to account for those factors which we did not, or could not, measure, while being pragmatic about the number of criteria we asked experts to rate on. While others indicator appraisal tools, like the Appraisal of Indicators through Research and Evaluation instrument [49] and the Quality Indicator Critical Appraisal Tool [47], include several attributes to assess indicators, they would likely be very time-consuming when considering a large number of indicators.

One of the limitations of our study was that we did not include indicators measuring all aspects of immunisation programs. This was a pragmatic consideration, as we acknowledged that experts were time-poor and unlikely to dedicate several hours to rate upwards of 100 indicators for the purpose of research. Even with fewer indicators, some experts noted that it was time-consuming to think through indicators using multiple criteria. Conducting the elicitation remotely possibly added to this challenge, especially as Pacific culture strongly values interpersonal and community connections [50] which were more challenging to establish through online meetings, whereas a face-to-face elicitation may have provided greater opportunity build rapport with experts and clarify experts' questions about the study including specific aspects of the instrument. If replicating this exercise for programmatic purposes and if resources permit, a half-day or full-day in-person workshop to work through a larger set of immunisation indicators would be ideal, as has been done in the past for other purposes [27, 51].

Our study included 13 purposively-sampled experts, which is arguably small to represent the diversity of countries, priorities and roles across different PICs, and may not be representative of all PICs. We made a pragmatic decision not to include experts from Papua New Guinea in this study as we considered the economic and political situation there to be vastly different to that of other PICs, noting our initial aim was to develop a single list of immunisation performance indicators for all PICs. However, there are many shared sociocultural determinants and operational challenges to delivering immunisation programs, and some of our experts referred to Papua New Guinea in discussions. Additionally, our method of identifying potential experts and recruitment via email and through professional contacts may have introduced bias into the study. Recruitment occurred while the COVID-19 pandemic response was ongoing, so some experts with experience highly relevant to this study may not have had capacity to participate in the study. Non-participation of some experts, including non-responders and experts who began but did not complete the study, meant that some countries may not have been adequately represented in this study. Nevertheless, included experts collectively had several years' worth of experience, with four individually having more than ten years' experience working in PICs. Experts also generally agreed that the highest ranked indicators reflected the most important immunisation priorities in their specific settings and across the Pacific, and the themes that arose in discussions were recurring, providing a degree of confidence in the validity of our results. We also acknowledge that our prior experiences, assumptions and beliefs may have influenced the research process, particularly the facilitation and analysis of discussions with experts. We attempted to minimise bias by providing all experts with an

opportunity to review their ratings and make changes, and having two authors independently review discussion transcripts to identify and categorise themes which were agreed to by all authors.

## Conclusions

Our study highlights the value of selecting performance indicators to monitor immunisation programs that are context-specific and aligned with the priorities, goals and capabilities of the country. Immunisation and health systems in the Pacific Islands and the challenges to immunisation programs are vastly different to those in other countries, and our study shows that differences even within the Pacific region mean that it would be inappropriate to have a single set of indicators measuring immunisation performance for all PICs. Rather, performance indicators should be country-specific, with countries having the flexibility to set short, medium and long term goals for their own immunisation programs, informed by evidence and their specific contexts. Further research is necessary to determine if this contextualised approach to monitoring and evaluation leads to tangible improvements in immunisation performance.

## Supporting information

**S1 Checklist. Inclusivity in global research.**
(DOCX)

**S1 Appendix. Expert elicitation instrument.**
(DOCX)

**S2 Appendix. Complete list of indicators included in the expert elicitation instrument.**
(DOCX)

**S3 Appendix. Statistical analysis of feasibility, relevance and preference scores, and ranking method.**
(DOCX)

**S4 Appendix. Definitions of constructs and themes identified through discussions with expert elicitation participants.**
(DOCX)

**S5 Appendix. Feasibility–proportion of experts who believed data for the indicator was collected, reported and of high quality, and mean feasibility scores, by indicator.**
(DOCX)

**S6 Appendix. Mean crude and weighted relevance scores and confidence scores, by indicator.**
(DOCX)

**S7 Appendix. Ten highest ranked indicators, by differing rating criteria and ranking method (results of sensitivity analysis).**
(DOCX)

**S8 Appendix. Ten highest ranked indicators, by demographic and professional groups.**
(DOCX)

**S9 Appendix. Relevance versus feasibility scores for indicators, by demographic and professional groups.**
(DOCX)

## Acknowledgments

We would like to thank all experts who participated in this study for the time and contributions. We would also like to thank Dr Md. Shafiqul Hossain, Dr Kerrie Wiley and Ms Nicole Rendell for testing and piloting the elicitation instrument.

## Author Contributions

**Conceptualization:** Cyra Patel, Ginny M. Sargent, Akeem Ali, Meru Sheel.

**Data curation:** Cyra Patel.

**Formal analysis:** Cyra Patel, Ginny M. Sargent, Adeline Tinessia, Dan Chateau, Meru Sheel.

**Investigation:** Cyra Patel.

**Methodology:** Cyra Patel, Ginny M. Sargent, Helen Mayfield, Dan Chateau, Meru Sheel.

**Project administration:** Cyra Patel.

**Supervision:** Ginny M. Sargent, Meru Sheel.

**Validation:** Ilisapeci Tuibeqa.

**Writing – original draft:** Cyra Patel.

**Writing – review & editing:** Cyra Patel, Ginny M. Sargent, Adeline Tinessia, Helen Mayfield, Dan Chateau, Akeem Ali, Ilisapeci Tuibeqa, Meru Sheel.

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
