## [Decision Letter · Decision Letter 0]

23 Apr 2024

PGPH-D-24-00512

Measuring what matters: context-specific indicators for assessing immunisation performance in Pacific Island Countries and Areas

Dear Dr. Patel,

Thank you for submitting your manuscript to PLOS Global Public Health. After careful consideration, we feel that it has merit but does not fully meet PLOS Global Public Health’s publication criteria as it currently stands. Therefore, we invite you to submit a revised version of the manuscript that addresses the points raised during the review process.

We look forward to receiving your revised manuscript.

Kind regards,

Oghenebrume Wariri, MD

Academic Editor

Journal Requirements:

2. Please include a complete copy of PLOS’ questionnaire on inclusivity in global research in your revised manuscript. Our policy for research in this area aims to improve transparency in the reporting of research performed outside of researchers’ own country or community. The policy applies to researchers who have travelled to a different country to conduct research, research with Indigenous populations or their lands, and research on cultural artefacts. The questionnaire can also be requested at the journal’s discretion for any other submissions, even if these conditions are not met.  Please find more information on the policy and a link to download a blank copy of the questionnaire here: https://journals.plos.org/globalpublichealth/s/best-practices-in-research-reporting. Please upload a completed version of your questionnaire as Supporting Information when you resubmit your manuscript.

Additional Editor Comments (if provided):

Reviewers' comments:

Reviewer's Responses to Questions

**Comments to the Author**

1. Does this manuscript meet PLOS Global Public Health’s publication criteria? Is the manuscript technically sound, and do the data support the conclusions? The manuscript must describe methodologically and ethically rigorous research with conclusions that are appropriately drawn based on the data presented.

Reviewer #1: Yes

Reviewer #2: Yes

2. Has the statistical analysis been performed appropriately and rigorously?

Reviewer #1: I don't know

Reviewer #2: Yes

3. Have the authors made all data underlying the findings in their manuscript fully available (please refer to the Data Availability Statement at the start of the manuscript PDF file)?

Reviewer #1: Yes

Reviewer #2: Yes

4. Is the manuscript presented in an intelligible fashion and written in standard English?

Reviewer #1: Yes

Reviewer #2: Yes

5. Review Comments to the Author

Reviewer #1: Summary: This paper outlines findings from a series of expert consultations conducted with the objective to provide insight on the utility of various immunisation system performance indicators for decision-making. The study authors should be commended for their interesting research question and appreciation of variation in needs across Pacific Island countries (PICs). This paper, however, could benefit from a more thorough discussion of the specific types of decisions that these personnel are making across their respective settings and responsibilities and how these types of decisions may influence their feedback on the suggested indicators.

Major comments:

1. Useful indicators across any setting and participant (given their affiliation) are likely to differ based on the underlying decision that is being made. Can you speak to the variation in decisions that are being considered across PICs (e.g., planning tailored campaigns by geographically or age, considerations for new vaccine introduction, etc.) and how this influences the relevance of the various indicators considered? This is likely the largest underlying driver of responses and was barely touched on.

2. To increase transparency, as well as provide more robust context for the expertise provided, study authors could provide the names, countries and affiliations of the experts consulted per this investigation somewhere in the report (e.g., as co-authors, acknowledgements, in the supplement).

a. Also, study authors should consider adding a section or information on how many participants were from countries with populations < 100,000 or some other metric.

b. Are countries with now coverage uniformly nationally versus those with substantial variation more likely to prefer certain indicators? Study authors mention this theme starting on line 450, but it feels incomplete and non-substantive.

c. Study authors illude to some of this information possibly being contained within the appendix (line 294)., but it is buried among a lot of information and needs to be more clearly highlighted.

3. Without adding information on which indicator is which to each point in figures 2-3, they aren’t actually very informative or interesting to the reader. At minimum, study authors could colour the points instead by indicator category instead of top/middle/bottom thirds (which is easy to see visually without colour indication).

Minor comments:

1. Line 56. Perhaps give a few examples of “indicators” to paint a clearer image for the reader. Something like “These immunisation-based indicators include more traditional metrics such as MCV1 coverage and DTP1-3 dropout and other less common metrics such as XXXX.”

2. Lines 79-83. Can you say anything yet about the quality of data from these electronic immunisation registers? Given many underlying biases of vaccination coverage administrative data, are these systems likely experiencing similar (or different) challenges regarding representativeness, completeness, etc?

3. The term “stakeholder” is used multiple times throughout the manuscript. Study authors should reflect on the use of this term and how it might perpetuate colonialism within the broader global health field. Please consider a more mindful term, such as “affected parties” or “decision-makers”. See https://www.cdc.gov/healthcommunication/Preferred_Terms.html for additional examples.

4. Are the methods for “preference” score validated? Or borrowed from other research? It would be helpful to know if study authors crafted this method themselves or if there is any validation of these methods? If independently derived, have study authors done any sensitivity analyses on methodologic choices?

a. On average, how many indicators did each expert choose to keep versus remove? This would give some more context to the quantitative preference scores that are otherwise difficult to interpret.

Reviewer #2: Thank you for the opportunity to review this manuscript which reports the findings of an expert elicitation study conducted to better understand how experts prioritise indicators of national immunisation performance in Pacific Island Countries and Areas. The outcomes of this study are quite interesting and should help inform global partners on how best to tailor indicators for settings with varying contexts. Overall, the manuscript is very well written, the methodology outlined is sound, and the implications drawn are relevant for the field. I have the following minor comments for the authors’ attention.

• Line 81, page 4; “During the COVID-19, some PICs introduced new digital systems…” Kindly insert “pandemic” after “COVID-19”.

• Line 82, page 4; “…to collect and aggregate immunisation data pandemic, and are at varying stages…” Kindly review this statement and rephrase appropriately for ease of clarity.

• Lines 158 – 159, page 9; “…minor revisions, the elicitation instrument was finalised.” For transparency and reproducibility, kindly expand on what amendments were made to the instrument after the piloting process.

• Line 165, page 9; “Experts then independently completed the elicitation instrument…” The authors should indicate in what language(s) the instrument was administered and whether translation / back translation was required for the purpose of analysis. This should also be clarified with regards to the qualitative discussions.

• Line 188 – 189, page 10; “Informed consent was obtained verbally during the first meeting.” The authors should expand on this, indicating why written informed consent was not required/appropriate for a study of this design.

• Line 227, page 11; “Twenty-two experts agreed to participate in the study…” Kindly provide clarity on non-responders or participants who did not complete the study. Furthermore, based on their roles, representation, and characteristics, it will also be important to understand within the study limitations section whether those who did not respond after agreeing to participate or did not complete the study may have contributed any further significant findings to this study than what was captured from the 13 experts.

• Line 486, page 26; “…were asked about their preferences were for characteristics of a set of indicators…” This sentence is unclear and needs rephrasing.

• Lines 492 – 493, page 27; “Experts said fewer insights to be more likely to be reviewed and used than long lists of insights…” Kindly rephrase this statement for ease of clarity.

• Line 594, page 32; “…may have provided greater opportunity build connections and clarify questions.” Kindly rephrase for clarity.

6. PLOS authors have the option to publish the peer review history of their article (what does this mean?). If published, this will include your full peer review and any attached files.

**Do you want your identity to be public for this peer review?** For information about this choice, including consent withdrawal, please see our Privacy Policy.

Reviewer #1: No

Reviewer #2: **Yes: **Edina Amponsah-Dacosta

---

## [Decision Letter · Decision Letter 1]

21 Jun 2024

Measuring what matters: context-specific indicators for assessing immunisation performance in Pacific Island Countries and Areas

PGPH-D-24-00512R1

Dear Ms Patel,

We are pleased to inform you that your manuscript 'Measuring what matters: context-specific indicators for assessing immunisation performance in Pacific Island Countries and Areas' has been provisionally accepted for publication in PLOS Global Public Health.

Best regards,

Oghenebrume Wariri, MD PhD

Academic Editor

Reviewer Comments (if any, and for reference):

Reviewer's Responses to Questions

**Comments to the Author**

1. If the authors have adequately addressed your comments raised in a previous round of review and you feel that this manuscript is now acceptable for publication, you may indicate that here to bypass the “Comments to the Author” section, enter your conflict of interest statement in the “Confidential to Editor” section, and submit your "Accept" recommendation.

Reviewer #1: All comments have been addressed

Reviewer #2: All comments have been addressed

2. Does this manuscript meet PLOS Global Public Health’s publication criteria? Is the manuscript technically sound, and do the data support the conclusions? The manuscript must describe methodologically and ethically rigorous research with conclusions that are appropriately drawn based on the data presented.

Reviewer #1: Yes

Reviewer #2: Yes

3. Has the statistical analysis been performed appropriately and rigorously?

Reviewer #1: Yes

Reviewer #2: N/A

4. Have the authors made all data underlying the findings in their manuscript fully available (please refer to the Data Availability Statement at the start of the manuscript PDF file)?

Reviewer #1: Yes

Reviewer #2: Yes

5. Is the manuscript presented in an intelligible fashion and written in standard English?

Reviewer #1: Yes

Reviewer #2: Yes

6. Review Comments to the Author

Reviewer #1: (No Response)

Reviewer #2: Thank you for taking the time to address the comments raised in the first round of peer-review. Please do note however that some of the phrases I flagged for editorial editing and clarity still need rephrasing.

7. PLOS authors have the option to publish the peer review history of their article (what does this mean?). If published, this will include your full peer review and any attached files.

**Do you want your identity to be public for this peer review?** For information about this choice, including consent withdrawal, please see our Privacy Policy.

Reviewer #1: No

Reviewer #2: **Yes: **Edina Amponsah-Dacosta
